# Three-Dimensional Morphology and Size Measurement of High-Temperature Metal Components Based on Machine Vision Technology: A Review

**DOI:** 10.3390/s21144680

**Published:** 2021-07-08

**Authors:** Xin Wen, Jingpeng Wang, Guangyu Zhang, Lianqiang Niu

**Affiliations:** 1School of Software, Shenyang University of Technology, Shenyang 110870, China; guangyuz@smail.sut.edu.cn (G.Z.); niulq@sut.edu.cn (L.N.); 2School of Mechanical Engineering & Automation, Northeastern University, Shenyang 110819, China; 1970194@stu.neu.edu.cn

**Keywords:** machine vision, three-dimensional morphology, high-temperature metal components

## Abstract

The three-dimensional (3D) size and morphology of high-temperature metal components need to be measured in real time during manufacturing processes, such as forging and rolling. Since the surface temperature of a metal component is very high during the forming and manufacturing process, manually measuring the size of a metal component at a close distance is difficult; hence, a non-contact measurement technology is required to complete the measurement. Recently, machine vision technology has been developed, which is a non-contact measurement technology that only needs to capture multiple images of a measured object to obtain the 3D size and morphology information, and this technology can be used in some extreme conditions. Machine vision technology has been widely used in industrial, agricultural, military and other fields, especially fields involving various high-temperature metal components. This paper provides a comprehensive review of the application of machine vision technology in measuring the 3D size and morphology of high-temperature metal components. Furthermore, according to the principle and method of measuring equipment structures, this review highlights two aspects in detail: laser scanning measurement and multi-view stereo vision technology. Special attention is paid to each method through comparisons and analyses to provide essential technical references for subsequent researchers.

## 1. Introduction

High-temperature metal components, including shaft forgings, ring shell forgings, high-temperature rails, metal slabs and other hot forming parts, are an integral part of industrial production, aerospace equipment, national defense military equipment, and other fields. The continuing development of forming manufacturing technology has effectively improved the quality of metal components, but avoiding surface defects and dimensional errors on metal components is difficult. Real-time measurement of the key parameters of high-temperature metal components can ensure the product quality requirements, reduce the scrap rate, and obtain better economic benefits.

Since the measured surface temperature of high-temperature metal components can reach above 800 °C, manual measurement methods with simple heavy tools cannot guarantee measurement accuracy. The low measurement efficiency after cooling cannot meet production demands.

According to whether the measuring equipment is in contact with a measured object, 3D morphology measurement methods can be divided into contact methods and non-contact methods. A coordinate measuring machine (CMM) is a typical example of contact measurement equipment. However, the measuring efficiency of this kind of equipment is very low, and it cannot be applied to the measurement of high-temperature objects in complex environments.

Non-contact 3D morphology measurement is a measurement method based on photoelectric and electromagnetic technology that can obtain the surface morphology information of an object without touching the surface of the object. The equipment used in this measurement method does not contact the surface of the sample that is being measured and does not cause permanent damage to the surface of the sample; the method has the advantages of fast measurement speeds, high precision, high resolution and large measurement ranges. This method is widely used in various complex environments for 3D morphology measurement tasks.

Non-contact 3D morphology measurement methods include:Acoustic methods, such as 3D ultrasound imaging technology, have a wide range of applications in the medical field [1,2,3].X-ray detection methods, such as X-ray 3D imaging detection, can directly characterize the 3D morphology of a blocked part of an object [4,5,6,7].Optical methods include the optical probe method [8] and optical interference method [9].Machine vision methods, which use non-contact measurement technology, only need to collect multiple images of a measured object to obtain its 3D morphology and size data; hence, they can realize the 3D morphology and size measurement tasks of high-temperature metal components. These methods are technical methods that use machines to replace human eyes for measurement and judgment.

However, in the process of measuring the 3D size and morphology of high-temperature metal components, machine vision usually faces the following difficulties and challenges:High-temperature radiation can cause objects to transfer heat to the outside world and radiate visible light. As the temperature increases, an object can produce bright white light. High temperatures can cause the light to bend during propagation, which leads to unstable images, large deviations and even the inability to obtain images.Harsh working conditions. There exists a vibration influence in the actual production process of high-temperature metal components, which results in the deviation of measurement data.The presence of oxidation reactions on metal surfaces. Metal surfaces react with oxygen at high temperatures, producing a large amount of oxide coating and emitting banded light waves below 400 nm in the ultraviolet spectrum.

To solve the above difficulties and challenges, in recent decades, researchers have constructed a variety of measuring devices to measure the 3D size and morphology of high-temperature metal components. This paper summarizes the solutions to the difficulties and challenges in the measurement process of high-temperature metal components. Firstly, to reduce the influence of high-temperature radiation on the measurement results, a photography device with a filter can be used for data acquisition [10]. In addition, image processing techniques and image algorithms can be used to reduce noise and optical distortion errors. It is also a feasible solution to use an infrared thermal imager instead of visible light for measurement. Secondly, for systems working in harsh working environments, cooling systems such as water cooling or air cooling can be adopted to reduce temperatures, and internal air purification devices can be used to avoid harm to the equipment caused by dust accumulation in the factory environment. A shock absorber device can be added to reduce the effect of the actual production environment on precision. Thirdly, for the influence of the oxide sheet, a high-pressure purge device can be used to remove most of the oxide sheet before detection. Then, different robust image processing algorithms can be used to reduce the interference of oxide sheet and reduce the false detection rate. The specific experimental hardware equipment and related measurement principles will be given in detail in this paper.

With respect to the principles and method of construction of measuring equipment, this paper divides the methods into two categories: laser scanning measurement methods and multi-view stereo vision technology. Additionally, the characteristics of each method are compared and analyzed to provide an important technical reference for subsequent researchers. Figure 1 presents the development of various technologies and the improvement of equipment along the time axis.

The organizational structure of this paper is shown in Figure 2. Section 1 introduces the difficulties and challenges that machine vision technology faces in the measurement of high-temperature metal components. Section 2 summarizes the measurement methods of laser scanning. Section 3 reviews the measurement methods of multi-view stereo vision technology. Section 4 includes a discussion and comparative analysis. Section 5 looks forward to the future development trends. Section 6 presents the conclusion.

## 2. Laser Scanning Measurement Method

There are two basic principles of laser scanning measurement methods: the time-of-flight (TOF) principle and the laser triangulation principle. The application of these two principles in the measurement of high-temperature metal components is reviewed in the following section.

### 2.1. TOF

#### 2.1.1. Basic Principle

The basic principle of TOF, also known as the time-of-flight measurement [11,12,13], is shown in Figure 3. The system is composed of a laser emitting module and a receiving module. The laser pulse is continuously emitted to the high-temperature metal parts through the laser emitter. The pulse is reflected back when it touches the metal parts and is received by the sensor. The obtained distance information can be converted into the coordinate information of the feature points on the metal surface through certain algorithms to complete the measurement task.

#### 2.1.2. Introduction and Description of Specific Applications

Germany Minerals Technologies [14] developed a La Cam Forge system based on vibration and high-temperature conditions in steel production workshops with TOF range finding technology. The laser scanner used in the system can be installed near the forging due to its protective cooling shell and can obtain a good laser scanning field. The measurement range of this system includes the linearity, proper alignment and diameter of metal components. This system solves the online measurement problem of high-temperature forgings and can retain complete measurement data for finite element simulations. However, when the system is used to measure the metal components, a special gripper car is needed to control the movement of the metal components.

Zhisong Tian et al. [15] developed a large new forging measurement technology (laser range finder) based on pulse radar in 2009. The system consists of a pulsed TOF laser rangefinder, a 2-DOF spherical parallel mechanism single-point scanning device and two motors with controllers. In the measurement process, the point coordinates of a measured object are obtained by a continuous rotation of the scanner. The key technology of the system is the TOF and SPM scanning device, which consists of five space rods that obtain the coordinates of a point (ρ α β) by the sequential rotation and static motion of two motors; then, the coordinates of the point are converted into a rectangular coordinate system. The system only needs to obtain size information by fitting the morphology of the measured target in one measurement cycle. The length dimension can be obtained by two axial scans to determine the positions of the two end faces of a cylinder. Compared with other high-temperature metal component measuring equipment, the rigidity of the system is improved, and the deformation of the scanning device is reduced. The vibration and moment of inertia are reduced because the motor is mounted on the base platform of the system. However, the spherical parallel scanning mechanism is too complicated to be used on a large scale in forging workshops.

By testing the TOF system in a laboratory and forging plant, it was proven that the system can meet the measurement requirements of high-temperature forgings. The measurement error in the laboratory environment is 4.5 mm, and the measured length is always less than the actual length of the metal component.

In 2012, Jun He et al. [16] used a double PRRR robot system to measure a hot metal shell with sensors in a polar coordinate system. By adjusting the rotating platform to make the laser scanning plane perpendicular to the cylindrical axis, the system can adopt the circular fitting method to make the measurement results more accurate. The point coordinates of the target object are obtained through a continuous rotation of the laser measurement sensor controlled by the scanning device, and the Denavit-Hartenberg (DH) method is adopted to convert the coordinates of the measurement points into the same coordinate system. To ensure the vertical relation of the axis in the measuring mechanism and avoid the machining error and assembly error of the equipment, the motion parameters of the DH matrix are optimized by the joint motion trajectory method. The parameters of the DH matrix are adjusted several times to keep the rotation axis of the sensor parallel to the axis of the metal housing. In data processing, the team used geometric fitting to make the measurements more reliable. The measurement error percentages of the left and right cylinder end face diameters in the thermal state are below 0.235% and 0.205%, respectively. The PRRR scanning device is used to replace the complex parallel spherical scanning mechanism and simplifies the measurement process. However, the motion parameters should be adjusted several times during the measurement process to ensure the precise control of a certain space position of the mechanism.

In 2012, Bokhabrine et al. [17] used the time-of-flight measurement principle and equipped two Leica ScanStation2 time-of-flight laser scanners to measure the diameter of a cylindrical metal component in a high-temperature state. Simulation software based on the genetic algorithm was developed to determine the location of the laser scanner to solve the problem of the scanner’s perspective being blocked in the measuring environment. The collection process needs to rotate metal component by 120°, and a complete point cloud can be obtained after three collections. Bokhabrine adopted a 3D segmentation method to separate the 3D points of a metal component from the background and determine the direction of the cylinder axis under the point cloud Gaussian image. Finally, the ICP algorithm was adopted to reconstruct the point cloud. The measurement system was first experimentally measured on a cooled metal component and then applied to the production line measurement of a hot cylindrical metal component. According to the test results of high-temperature metal components, the system has high accuracy, and the measurement error is less than 8 mm. The measurement speed is greatly improved compared with the traditional measurement method. However, the measurement object can only be circular; otherwise, the algorithm for obtaining the point cloud fails, which limits its application scope. Moreover, it is difficult to ensure perfect coincidence between the marker and the spot when the acquisition process is rotated.

In 2016, the research team of Zhengchun Du [18] and Shanghai Heavy Machinery Factory Co., Ltd. (Shanghai China) studied the extraction and measurement of point clouds of high-temperature objects and proposed a solution for obtaining the forging profile in a complex environment. The radar device of the system is connected to the reducer and motor and is controlled by the motion control card. The scanning plane of the radar is perpendicular to the main shaft of the motor, which can realize the 3D space measurement of high-temperature objects and represent the measurement points in the polar coordinate system. Zhengchun Du et al. analyzed the distribution characteristics of point clouds in the horizontal and vertical directions and extracted point clouds of forgings. Step 1: Use the angle constraint and distance constraint conditions to roughly remove irrelevant points. Step 2: Calculate the curvature to extract the boundary points and further remove the irrelevant data in the environment. Step 3: Use hierarchical clustering analysis to remove false boundary points and obtain point clouds without interference information. After the point cloud has been successfully extracted, the 3D Delaunay triangulation method is used to reconstruct the model to realize the dimension measurement of high-temperature metal components.

The measuring time of the system for 3-m-long high-temperature metal components is approximately 15 s, and the dimensional error is less than 2%.

#### 2.1.3. Comparative Analysis

Based on the time-of-flight principle, a dimension and morphology measurement system for high-temperature metal components is developed. The measurement objects are mostly cylindrical shell metal components. However, the hardware equipment of the system is different, which is mainly due to the different working environments. The performance improvement in the system lies in the simplification of the scanning device and in the improvement in the reconstruction technology of target metal components, which accurately extracts the metal components in the measurement environment. Table 1 compares and analyzes the research results of the time-of-flight method from the aspects of the measurement target, measurement equipment and key technologies.

### 2.2. Laser Triangulation Method

#### 2.2.1. Basic Principle

The 3D information obtained by laser triangulation technology has the characteristics of high resolution and high accuracy [19,20,21,22], and its basic principle is shown in Figure 4. A laser triangulation system usually consists of a laser transmitter and a camera. The laser emitter emits point laser or line laser to the metal workpiece in the high temperature state to be measured from a certain angle, and then the CCD photodetector receives the laser spot or stripe on the surface of the metal part. The change of the size and morphology of the measured object can be reflected by the position change of laser spot and fringe captured by the camera.

#### 2.2.2. Introduction to Specific Applications

Researchers at Yanshan University have performed much work on the measurement of ring forgings [23,24]. In 2014, the research team of Professor Yucun Zhang [25] used the principle of laser triangulation to measure the diameter of high-temperature cylindrical forgings. The research team conducted an in-depth study on system calibration using the PSO algorithm to optimize the internal and external parameters of the camera to improve the accuracy of system measurements. With a rectangular block of a known height, the laser scanning plane is solved through the corresponding relationship between the points on the image and the spatial points. The results of the system were verified with an average error of less than 0.25 pixels in the vertical direction and 0.5 pixels in the horizontal direction. Finally, the obtained scan points are converted to the same world coordinate system, and the least square method is used to fit the contour of the measured object to achieve 3D measurement with a measurement error of less than 1 mm.

The research team of Professor Yucun Zhang [26] has made continuous breakthroughs in the research of high-temperature metal components. They proposed a detection method for cylindrical forgings in a rotating state with a rotation speed of 0.3 m/s in 2017. With the scanner made by the Riegl company of Austria to scan the same section of forgings many times, the distance between the scanner and forgings is obtained and converted into two-dimensional (2D) coordinate information. A threshold is set based on the size of the distance to remove coordinate points that do not belong to the forging. In the aspect of forging contour reconstruction, the least squares method is also used to fit the circle. Since the forging is rotating, the scanned data also need to be rotated. The system was tested by the China First Heavy Industry Group, and the measurement error of the outer diameter was less than 6 mm. However, a sufficiently large scanning range is required to ensure that the same section is scanned during measurement, so there are limitations in practical applications.

In 2018, Professor Yucun Zhang’s research team [27] carried out measurements on complex annular forgings with free-form surfaces. The system takes a top-down approach to laser scanning to obtain contour point cloud data from forgings. As the diameter of the ring forging increases with height, the extracted contour line shifts positions. Therefore, the Kalman time-sharing multiplexing particle filter method is used to dynamically track the noise and error and to perform error compensation on the extracted forging point cloud data. With topology and information system quadruple theory, a space model of ring forgings is built, and a model with a smooth surface is finally obtained through topological embedding mapping. Topological differential theory builds a tangent cluster model and completes the extraction of surface features. The system measures the height of each step of ring forging and has higher accuracy compared with other algorithms, and the flatness deviation is less than 1.74 mm.

In 2015, Alexander Schöch et al. [28] developed a system (CMS) that can perform 3D measurements of high-temperature objects with complex surfaces with the support of a hot-gauge project. Eight sensors composed of a laser emitter and an industrial CMOS camera are arranged in octagonal morphology to measure the 2D section of an object. The information in the direction of the third coordinate is obtained by moving the platform. To protect the camera and measuring frame, the system is equipped with an air cooler and water-cooling pipes to control the temperature, as well as protective devices such as dust control equipment and protective panels. To verify the performance of the system, measurement experiments were performed. The length measurement error range is 0.1 mm, and it takes approximately 4.2 s to measure a 700 mm turbine blade. However, due to the arrangement of the system sensors, only two-way measurements can be carried out.

In 2016, Veitch-Michaelis et al. [29] combined the principle of laser triangulation with machine learning to detect surface defects on cast steel plates. This project (high-temperature process control (HTP-C)) uses a striped laser transmitter that can emit 405 nm. Compared with the traditional triangulation system, a camera on the same axis as the laser beam is attached to the laser beam launch port. Due to the high temperature of the measurement environment, the system uses a blue filter and cooling housing to reduce the environmental impact. The center of mass peak detection algorithm is used to detect the laser stripes, and the laser line can be accurately extracted during the casting process of the metal component. Defect detection is divided into two stages. The first stage adopts image processing algorithms, such as morphological edge detection and median filtering, to obtain the detection results without considering false detection. The second stage uses a support vector machine (SVM) and training data to obtain an accurate defect detection result. As the width of the slab increases, the high temperature has an impact on the measurement system, causing the temperature of the laser transmitter to rise. When a 1970-mm-wide slab was tested, the system could not be used normally in most cases. An exposure time is also required for system measurements; otherwise, some small crack defects cannot be detected.

In 2017, Drago Bračun et al. [30] analyzed the interference of two factors, namely, the surface oxide scale on high-temperature metal components and the light source in the measurement environment on system measurements, and used the principle of differential imaging for measurement. The two images collected by the camera are used for one measurement; one image is the image radiated by the laser, and the other is not radiated by the laser. A clear laser contour image can be obtained from the difference between the two images. After median filtering is applied to the obtained difference image, the contour can be extracted, and the pixel information of the contour can be converted into Cartesian coordinates to obtain more accurate measurement values. The system calibration uses a triangle waveform as the calibration body and the triangle vertex as the calibration point to complete the comparison between the output point and the measurement point. Differential processing can obtain the laser fringes very well, and the measurement uncertainty is less than 1 mm, but the camera collects 50 images per second and only performs 25 measurements, which reduces the measurement speed.

The NextSense company [31] developed the CALIPRI RCX system using laser reconstruction methods, which can detect hot-rolled metal components. The system is an off-line measurement system, and measurements can be achieved by placing the measuring instrument at any position on the rolling mill. When the system is working, the sensor continuously rotates around the rolling piece to obtain cross-sectional contours in different directions to form an overall contour. Compared with other measurement systems, this system takes up less space and moves more flexibly and easily.

#### 2.2.3. Comparative Analysis

The hardware devices of the high-temperature metal component measurement system developed by the principle of laser triangulation include laser transmitters and cameras. For different measurement objects, some systems emit point lasers, and some systems emit line lasers. In addition to measuring cylindrical forgings, the system can also measure parts with complex surfaces. Different systems also differ greatly in measurement accuracy. Table 2 compares and analyzes the research results of laser triangulation from the aspects of the measurement target, measurement equipment and key technologies.

## 3. Multi-View Stereo Vision Measurement Method

### 3.1. Basic Principles

The multi-view stereo vision measurement method arranges cameras around a metal component to be tested, obtains target information from different perspectives, and then performs stereo matching to complete the measurement task based on the 3D model. The measurement method of multi-view stereo vision is based on monocular and binocular vision and can be composed of multiple monocular or binocular systems. The basic principle of binocular stereo vision is similar to the imaging system of the human eye [32,33,34] and consists of two cameras located in the same plane, with the optical axes parallel and at the same height.

Figure 5 shows the measurement principle. Based on the small hole imaging model, the mutual conversion between the feature point $P\left(X_{W},Y_{W},Z_{W}\right)$ of the spatial metal component and the corresponding image points $\left(x_{l},y_{l}\right)$ and $\left(x_{r},y_{r}\right)$ is completed in the world coordinate system, the camera coordinate system and the image coordinate system. In addition, camera calibration technology and 3D reconstruction technology are also needed.

At present, research on binocular stereo vision is often combined with structured light technology. By replacing a camera with a structured light emitter or directly adding a structured light emitter, a structured light binocular stereo vision system is established. The following is an overview of the research work in the field of stereo vision measurement of high-temperature metal components.

In multi-eye stereo vision measurements, the structured light 3D measurement method is the most commonly applied method and has been widely used in plant surface feature recognition, human body feature recognition, 3D sensing, cultural relic protection, metal sheet size control and medical fields [35,36,37,38,39,40,41,42]. How to effectively evaluate the measurement stability of many structured light 3D measuring instruments is one of the keys to improving the processing quality of manufacturing products. In 2019, Pooya Ghandali et al. [43] proposed a structured light technique for performance assessment in dimensional and topographical measurements. In the same year, George Gayton et al. [44] developed an evaluation framework for the uncertainty of the extraction of feature point clouds on the surface of objects to correct them in time.

However, for 3D measurements based on structured light, different coding methods affect the measurement accuracy of the system on the technical level [45,46]. In 2018, Song Zhang [47,48] conducted research on the application of structured light technology in 3D profile measurement, introduced in detail the imaging principles and system calibration methods of structured light technology, compared the influence of different coding methods on the experimental results, and deeply analyzed the challenges and development trends of structured light technology. The research results of structured light stereo vision in recent years are introduced in the following according to their different coding methods.

#### 3.1.1. Sinusoidal Phase Encoding

The measurement principle of grating projection is to project a coded fringe pattern onto the measured target; the pattern is modulated by the surface morphology of the object and received by a camera to obtain the phase through a coding and decoding process, and the depth information is reflected according to the phase. According to different calculation methods, fringe projection profilometry (FPP) can be divided into phase shifting profilometry (PSP) in the time domain and Fourier transform profilometry (FTP) in the frequency domain. Phase shifting profilometry is a digital grating projection technology and the most typical structured light 3D measurement technology. It uses the phase-shift principle to calculate the phase, and the three-step phase-shift formula is as follows:(1)$$I_{i}\left(x,y\right)=I^{\prime }\left(x,y\right)+I^{\prime \prime }\left(x,y\right)cos\left[\theta \left(x,y\right)+\alpha_{i}\right]$$
where *i* is the number of phase shifts, $I_{i}$ is the gray value of point (*x*, *y*) on the i-th phase-shifting image, $I^{\prime }\left(x,y\right)$ is the background gray value, $I^{\prime \prime }\left(x,y\right)$ is the amplitude of modulation, θ(x,y) is the phase to be calculated, and $\alpha_{i}$ is the phase shift value of the *i*-th phase-shifting image.

Next, the research status of sinusoidal phase encoding is summarized in chronological order. The development achievements in recent years are chronologically shown in Figure 6.

1.The research results of sinusoidal phase encoding obtained in 2018 are as follows:

Song Zhang [49] analyzed various factors, such as ambient light and noise, that affect the image quality of camera sensors and proposed the best exposure control technology for structured light fringe projection. The collected fringe image only needs to be passed once to determine the best global exposure time for 3D measurement and obtain the high dynamic range (HDR) exposure time. By building a hardware system and experimenting on eight plates with different reflectivities, the effectiveness of the method was verified. This technology provides high-quality image information for structured light phase-shift coding and can obtain more stable measurement data.

The recovery of the absolute phase is limited by the depth of the scene. Yidan Xing and Chenggen Quan [50] proposed a new phase unwrapping method (ERDF) for an accurate calculation of the absolute phase. This method only needs to calculate the phase based on two sets of fringe patterns.

2.The improvements in the measurement stability of the system obtained in 2019:

Junpeng Xue et al. [51] did not use a DLP projector as a fringe projection device, but rather a galvanometer scanner with a projection speed of up to 500 fps. Compared with the commonly used DLP system, it has a similar measurement accuracy.

Yuwei Wang et al. [52] proposed an enhanced fringe phase encoding method. A method in which a single codeword corresponds to a half cycle of a stripe is adopted, and constraints are used to increase the coding range.

Xinran Liu and Jonathan Kofman [53] improved Fourier transform profilometry (FTP). Two sinusoidal phase images with phase difference are subtracted to remove the background intensity, and the 3D morphology is reconstructed by the fringe pattern after background modulation and geometric constraints are applied.

Phase encoding requires a certain number of code words, which can produce artifacts when the phase unwraps. To solve this problem, Shenzhen Lv [54] et al. adopted a path tracking method to improve the retrieval quality of the absolute phase and extracted fringe patterns through image segmentation processing. This approach applies not only to phase coding but also to binary coding and gray coding.

3.The latest research progress in grating projection technology accomplished in 2020:

The fringe projection measurement method for surface diffuse reflection objects has been dramatically improved, and the 3D measurement of highly reflective objects relies on direct phase measuring deflectometry. Xiaohong Liu et al. [55] merged the above two methods and measured 3D objects with both diffuse reflection and specular reflection. The built 3D system is finely calibrated, but there are still some potential error factors that need to be further resolved.

Kai Liu et al. [56] divided the images into several groups with the same phase shift, calculated the phase size of each group, and then averaged the phase sizes to obtain the final phase. This method can improve the nonlinear distortion of structural optical systems.

Lei Lu et al. [57] identified the targets in a scene and found their bounding boxes. Kernelized correlation filters (KCF) are used to track the detected moving target, the motion state is described by the rotation matrix and translation matrix, and finally, 3D reconstruction is completed.

Cheng Fei et al. [58] used shortwave infrared cameras to monitor targets in low-brightness environments. Improved Fourier transform profilometry is used to obtain the depth information, and a plastic model is used in the experiment to evaluate the measurement accuracy of the system.

Konstantinos Falaggis et al. [59] used the advantages of gray level (GL) coding techniques and combined FPP and PMD techniques to obtain surface information. This method can expand 512 stripes by adding an RGB image.

4.Many innovative new technologies have been developed with other algorithms, such as deep learning:

In 2019, Haihua Zhang et al. [60] replaced the DLP projector with a homemade mechanical projection device, borrowed the advantages of FTP and PMP, and developed time Fourier transform profilometry (TFTP). This method mainly operates in the time domain and can measure independent dynamic objects. As a major unwrapping method, spatial phase unwrapping has certain computational instability. Sen Xiang of Wuhan University of Science and Technology and You Yang [61] developed a more robust phase unwrapping method called multi-anchor scanline unwrapping (MASU). With statistical methods, multiple pixels are used as anchor reference points to effectively deal with the error transmission phenomenon existing in the spatial phase. Wei Yin et al. [62,63] used a neural network in a supervised learning mode to perform phase unwrapping on a fringe projection (DL-TPU). Hieu Nguyen et al. [64] combined a convolutional neural network to develop a 3D reconstruction technique. Fringe projection profilometry (FPP) is used to generate a data set, which is used as the input of the neural network for training and verification. The neural networks used include the fully convolutional network (FCN), AEN and UNet. Jiashuo Shi et al. [65] took advantage of a deep neural network (DNN) in the phase expansion of fringe projections and used the DnCNN model to learn and train the fringe extraction process, which enhanced the phase recovery of a single-frame fringe image. Jiaming Qian et al. [66] developed a technology that can realize all-round 360° 3D reconstruction based on fringe projection profilometry and relieved the limitation of viewing the angle occlusion of traditional equipment. Through the method of coarse registration and then fine registration, point cloud matching is completed based on the ICP algorithm.

In 2020, Haotian Yu et al. [67] built a sine computing neural network (FPTNet). This network contains two subnetworks, FPTNet-C and FPTNet-U, which perform phase calculation and phase expansion, respectively. This method reduces the number of fringes needed to understand the phase. Congyi Lyu and other researchers [68] developed an embedded binocular structured light 3D measurement sensor with ARM and FPGA as the main computing unit. The PMP algorithm is divided into four phase calculation units, which can meet the requirements of real-time high-speed data measurements. Haihua An et al. [69] developed CGPMP technology based on the measurement accuracy of phase measurement profilometry (PMP). The influence of phase noise and amplitude noise on measurement accuracy is well suppressed by a hybrid filtering window.

5.The breakthroughs in system calibration and matching of 3D measurements with phase-shift profiling mainly include the following related research work:

In 2018, Wenjing Zhao et al. [70] adopted multi-step phase shift technology to collect the phase of structural light and calibrate each pixel. Compared with the traditional 3DSTC, the system reduced the influence of distortions.

In 2019, Rigoberto Juarez-Salazar and Victor H. Diaz-Ramirez [71] conducted research on structured light systems with multi-frequency phase shift coding. A yellow physical chessboard and a cyan projection chessboard are used to calibrate the structured light system, and the color information is used to detect the calibration board, which relieves the constraint on the position of the measuring equipment.

In 2019, Congying Sui et al. [72] projected multi-frequency phase-shifted sinusoidal fringes onto a measured object and developed a multi-step matching 3D model reconstruction framework. This method eliminates the steps of phase unwrapping and projector calibration and reduces the system cost.

In 2020, Raúl Vargas et al. [73] improved the system calibration method of fringe projection profilometry (FPP) and proposed a stereo calibration method. The reconstruction results of flat, cylindrical and complex curved objects proved that this new method is superior to other stereo-vision (SV) methods.

#### 3.1.2. Binary Defocusing

Digital binary defocus technology performs reasonable defocus processing on the projector of a system so that the projected fringe pattern changes. By projecting defocused fringes onto the surface of an object, higher precision measurement results can be occasionally achieved.

Phase unwrapping mainly includes spatial phase unwrapping (SPU) and time phase unwrapping (TPU). In 2019, Wei Yin et al. [74] adopted a depth constraint method to make high-frequency phase unwrapping more stable. The number theory method is applied to the binary defocus projection.

#### 3.1.3. Statistical Pattern

Statistical random coding can realize the miniaturization of a system, so it is widely used in some electronic devices. Compared with other coding strategies, 3D reconstruction can be achieved by projecting speckles onto a target. The progress in the research on statistical coding patterns is as follows:

In 2019, Wei Yin and Chao Zuo [75] used composite structured light fringes in which the speckle pattern was embedded in the phase-shift fringe for 3D measurement. A complete measurement technology framework is proposed. Through algorithm optimization and the regional diffusion compensation technique (RDC), the periodic error of the package phase can be eliminated, and the mismatched area can be corrected.

#### 3.1.4. Binary Coding

With the continuous application of statistical coding methods, disadvantages have gradually been discovered, so scientific researchers have proposed binary coding. Gray code is a commonly used binary code. Two consecutive values have only one binary difference. The encoding and decoding process is as follows:(2)$$G_{i}=B_{i}⊕B_{i+1}\left(0\leq i\leq n-1\right)$$
(3)$$B_{i}=G_{i}⊕B_{i+1}\left(0\leq i\leq n-1\right)$$
where *G* represents Gray code, and *B* represents binary code.

In 2019, Guijin Wang et al. [76] conducted research on 3D object reconstruction in scenes with great depth. The method is divided into four steps: local imaging, fringe extraction, estimation of global depth values and 3D fusion. Compared with other methods, the results of 3D morphology are clearer and more comprehensive.

Zhoujie Wu et al. performed much work on gray coding for 3D high-speed measurements [77]. They established a 3D object morphology modeling system based on cyclic complementary gray code (CCGC) patterns [78]. The frequency of the projected sinusoidal fringes remained the same, and the phase shift algorithm and complementary gray code (CGC) were used to calculate and expand the phase. In contrast to experiments with traditional methods, this cyclically coded fringe pattern expanded the phase range and obtained more accurate experimental data.

The combination of the phase shift algorithm and gray code is widely used, and the factors that cause phase errors are also increasing. Zhoujie Wu et al. [79] proposed a shifting gray code (SGC) coding strategy to improve the calculation accuracy of a phase. By measuring the radius of a ceramic sphere, the effectiveness of the system was verified, and the performance was better than that of the CCGC mode.

In 2020, Yuping Ye et al. [80] developed an infrared structured light sensing system by adopting a gray code and phase shift coding strategy. The calibration of the IR SLS system measurement perspective was completed, and parallel computing was realized through CUDA in data processing.

#### 3.1.5. Other Research Work

In 2019, Shijie Feng et al. [81] used deep learning and proposed a new 3D morphology reconstruction method called micro deep learning profiling. The use of a neural network to extract the phase improves the efficiency of 3D reconstruction, and the measurement accuracy is better than that of other encoding methods. In the same year, Feifei Gu et al. [82] adopted a method for the precise positioning of main features (MFs) and auxiliary features (AFs); the method increased the encoding density by three times and realized the high-density encoding of structured light technology. Line structured light often requires motion control equipment to scan the whole picture of the measured object.

As shown in Figure 7, structured light has been applied to different tasks, and the research work is summarized according to the coding method and time.

#### 3.1.6. Analysis of Structured Light Technology

Most current structured light 3D systems use one camera to obtain images, and up to four cameras can be used. In the past three years, much work has been completed to improve image quality, phase acquisition accuracy, dynamic target detection and detection of objects with high reflectivity. The structured light technology of different encoding methods is compared and analyzed in the form of a table. Table 3, Table 4, Table 5, Table 6 and Table 7 compare the structured light technology according to different coding methods.

### 3.2. Specific Application Description

In 2001, Aceralia and the University of Oviedo developed a hot slab surface inspection system under the funding of the European Union’s ECSC steel plan [83]. Two of the subsystems can realize online recognition of surface cracks and inclusions in the slab. Surface crack detection adopts the method of fusing the gray image data of the CCD linear camera and the surface profile data based on cone holography to determine the location of a crack. Internal defect detection is carried out on the edge of the slab, and the CCD area camera is used for detection after removing the surface. The inclusion situation of the entire slab is estimated based on the detection results at the edge of the slab. Finally, the processing quality of the slab is comprehensively evaluated according to the test results, and subsequent processing methods are designed. The system was tested online in the company’s continuous casting steel plant, and the reliability of defect detection is above 90%. However, the system needs to remove a certain surface to achieve internal defect detection; thus, the moving speed of the metal component cannot be fast, which limits the detection efficiency. Moreover, the system uses many types of sensors, and the data processing process is cumbersome.

In 2003, the OG Corporation and several American universities and research institutes developed the HotEye^®^RSB system for reconstructing the morphology of high-temperature metal components under a cooperative project supported by the US Department of Energy using binocular vision principles [84]. The system is composed of a sensor module, a computer module and a cooling module, which can detect defects in hot-rolled metal components online to find problems in the production process in time, reduce the scrap rate and improve production quality. The HotEye^®^RSB system can measure the temperature of metal components up to 1450 °C. However, it can only measure a small metal component with a size of less than 100 mm, and the measuring range is small.

In 2007, the Kobe Steel Corporation also adopted the binocular vision principle to construct a system for reconstructing large high-temperature metal components [85]. The left and right cameras of the system are installed at a distance of approximately 20 m from the metal component of the high-temperature circular shell. Auxiliary measurements are not required by active irradiation, and the inner and outer diameters, thickness and axial length of a cylinder can be measured. Researchers use infrared light for optical calibration to obtain the optical parameters of the system. The system was verified in a laboratory and has a measurement accuracy of ±5 mm or less, which is approximately a ±10 mm difference from the accuracy of traditional measurement methods.

In 2011, the research team led by Professor Yuchi Lin [86] used an infrared thermal imager to form a binocular stereo vision measurement system. The system includes an infrared lens, thermal imager, optical platform, position control device, image acquisition system and image processing computer. To obtain the internal and external parameters of the measurement system, the infrared imaging system needs to be calibrated. One of the infrared thermal imaging camera positions is taken as the world coordinate system, which simplifies the system calibration process. Multiple matching algorithms of epipolar constraints and relaxation constraints are used to complete binocular stereo vision matching. The advantage of infrared thermal imaging cameras is that they can overcome the heat radiation of high-temperature objects. Compared with the visible light measurement method, it has higher accuracy. When measuring in the dusty environment of a factory, the measurement distance has little effect on the result. Infrared thermal imaging cameras have unevenness and are processed by an optimized artificial neural network correction method, but even after correction, the unevenness is not completely eliminated, and the system is still in the laboratory measurement stage. The experimental results show that the relative error of the system is less than 2%, and the standard deviation is less than 0.4 mm.

Zhelin Li et al. [87] developed a stereo vision system in 2013 to study how stereo vision systems can obtain clear image results. To solve the problem of high-temperature radiation from long-axis parts, the study found that the integration time affects the quality of the image obtained. A clear CCD image can be obtained by controlling the exposure time, and image noise can be reduced through a combined filtering algorithm. Then, the image segmentation algorithm is used to identify the target of a forging and distinguish the forging from the environment. Due to the processing characteristics of shaft parts, the CCD camera of the measurement system is arranged up and down, and checkerboard calibration is used to calibrate the system. Since the results of general edge detection algorithms cannot effectively extract the features of metal components, we have developed a method to obtain the gray distribution curve through cross-section lines, extract the feature points of shaft parts, and then match the extracted feature points through Euclidean distance. The accuracy of the system can reach 2.5–3.5 mm, and the measurement speed is five times per second. However, since the camera can only obtain a part of the cross-section of the component, more than half of the cross-section is blocked, and the blocked part needs to be obtained by reasoning, which increases the complexity of the system.

The research team of Professor Zhenyuan Jia and Wei Liu improved the machine vision inspection of large hot forgings [88,89,90,91]. With the support of the National 973 Program, the National Natural Science Foundation of China’s Scientific Instrument Fundamental Research Special Fund and other projects, in-depth research was conducted on the morphology and size monitoring of high-temperature metal components [92]. Due to the influence of the high temperature of the measured metal component, fitting an optical fringe with a Gaussian distribution will reduce the accuracy of extracting feature points. Therefore, the Weibull distribution model was developed to fit the fringes. After the light stripe model is established, the center point of the stripe needs to be extracted; the top hat transform is used to reduce the brightness of the background, and the image processing morphology operation removes the uninteresting picture area. Since the light stripes are well extracted, 3D matching is completed according to the intersection points by projecting vertical and horizontal stripes on forgings. This system also has the problem of measurement blindness. The measurement accuracy is related to the number of center points, and the relative error of the diameter measurement is less than 0.7%.

In 2016, a research team improved an image acquisition method [93], evaluated the image quality through the signal-to-noise ratio model of the region of interest, and analyzed the factors that affect the image quality. To meet the requirements of image clarity at high temperatures, ISNR was compensated by changing the laser power. The maximum absolute error is 0.32 mm, and the maximum relative error is 0.27%.

In 2015, based on the principle of binocular vision, a research team built a multi-camera stereo vision system [94]. Through the system calibration of each group of cameras, the feature points of large high-temperature metal components are converted to the same coordinate system. By controlling the exposure time of the CCD camera and distinguishing the laser fringe from the background, a clear fringe can be obtained. The measurement accuracy of this method can reach 0.10%.

In 2015, Xianling Zhao et al. found that infrared light emitted by some high-temperature objects interferes with PMP measurements [95]. Therefore, blue sine structured light was used to measure high-temperature objects (below 1200 °C). One of the CCD cameras in the binocular system was replaced by a DLP projector, and a 3D measurement system was formed with another 3-CCD camera. The 3-CCD camera and DLP projector were arranged at a certain angle to collect images of metal components. Then, the three channels of the image were separated to obtain the images under the R, G, and B channels. Phase wrapping and phase unwrapping were carried out with the obtained gray image under Channel B, and the unwrapping phase was corrected by a filtering algorithm. The measurement system can complete measurements within a few seconds, and the relative error is less than 1:1000. This system can obtain images with good sinusoidal characteristics, but a high-temperature environment has a great impact on hardware equipment, resulting in a measurement distance of less than 5 m.

In 2017, Yunhui Yan et al. applied structured light binocular measurements to the height measurement of high-temperature steel and established a 3D detection system [96]. Physical filters and digital filters are used to reduce the influence of surface high-temperature radiation from metal component at 1200 °C. The combined filtering technology uses blue physical filters and digital image processing technology to suppress the R component. Based on the problem of the long time requirements of gray coding calculations, the column coding mode is corrected by the principle of stereo correction. The CPU and GPU process image data in parallel; offline operations, such as system calibration and filtering, and stereo correction are performed by the CPU, and a matching operation after decoding is performed by the GPU, which improves the measurement speed. When testing the system in a laboratory environment, the minimum error was less than 1 mm, and the average error was less than 2 mm. At present, this measuring device measures static objects.

In 2017, Liya Han et al. [97,98,99] installed a binocular vision system with a structured light projector on an industrial robot to form a 3D measurement system for high-temperature metal components. Through the continuous movement of the structured light scanner driven by a robot, 3D data can be obtained from multiple viewing angles, which solves the limitations of environmental occlusion and a single viewing angle. An infrared cut filter is installed on the camera to avoid the influence of infrared rays on the image quality. Furthermore, to reduce the exposure time of the equipment in a high-temperature environment, the moving path of an industrial robot is designed. The transformation matrix is obtained by calibrating the scanner coordinate system, robot wrist coordinate system and robot base coordinate system, and the data can be converted to the same coordinate system. The included angle between the reference axis of the forging and the measuring axis is 0.0013 rad, and the distance is 0.28 mm.

To make full use of the bandwidth of Gigabit Ethernet, Liya Han et al. [100] in 2018 used a time division multiplexing trigger to divide the exposure time into two equal segments and performed exposure and data transmission, which improved efficiency. The repetitive characteristics of the industrial robot motion trajectory are used to align the 3D data obtained from different perspectives. The ICP algorithm is used, and the distance threshold is continuously updated during the measurement process to remove irrelevant background information and perform point cloud data fusion.

In 2019, Liya Han et al. [101] studied the method of vibration compensation in the measurement environment. Vibrations have a particularly obvious influence on the phase unwrapping process of the multi-frequency phase shift method, causing the image pixels to fail to correspond. Therefore, the vibration error compensation method based on a homography matrix is used to correct the pixel correspondence between package phase maps. When this algorithm is used to compare the sizes of a 3D reconstruction and an actual object, the average upper and lower deviations are +0.055 mm and −0.054 mm, respectively. The success rate of vibration compensation depends on the feature point extraction and matching algorithms. In addition, the system was developed for specific application scenarios and can continue to be improved for other scenarios.

In 2018, Yijun Zhou et al. [102] conducted research on binocular stereo vision measuring systems and built the system. In view of the large amount of data and long time needed for processing traditional characteristic point measurements, a characteristic line is innovatively used to replace characteristic points for the measurement of high-temperature forgings. The 3D contour line of an object surface is obtained through back projection of the characteristic line and the size of the contour line is obtained. The method of 3D reconstruction with characteristic lines avoids the influence of noise caused by characteristic points, does not require additional processing of image noise, and reduces the measurement cost. A 3D measurement experiment was performed on the gypsum model in a laboratory environment, and the contour line of the model was extracted by the Hough transform. Additionally, the measurement accuracy was basically the same as that of the feature point measurement, but the feature line measurement speed was better than the feature point measurement speed. In a forging workshop, a clear image was obtained by automatically adjusting the exposure time of the camera. The relative error of the measurement was 0.79%, and the measurement time was 1.9128 s.

In 2020, Liming Zhao et al. [103,104] combined deep learning with binocular stereo vision to establish an improved 3D laser image scanning system (3D-LDS). This system emits a green laser with a wavelength of 532 nm to reduce the influence of high-temperature radiation on casting and controls the integration time of the CCD to accurately obtain laser stripes. To build a deep CNN architecture that combines a fully connected network and a fully convolutional network, deep learning neural networks are applied to divide surface detection into two steps: defect recognition and description. The softmax function is used to perform multi-class training on the neural network to calculate the probability of a certain defect. The precise identification of a defect can reduce the redundant candidate bounding boxes generated by a wrong identification, and the advantages of the CNN are fully used. FP and FN evaluation indicators are used to evaluate the test results. As there are many types of defects, in-depth research can be conducted on the optimization of the network architecture to address more comprehensive inspection tasks.

### 3.3. Comparative Analysis

In recent years, in the measurement of various high-temperature metal components, multi-eye stereo vision technology has been one of the main research directions of scientific researchers. Systems developed by researchers have wide measurement ranges, which allows for the detection of metal slab defects and the measurement of the geometric dimensions of forgings and complex parts. Most of the equipment used is CCD industrial cameras and DLP projectors, and some researchers have applied infrared thermal imaging cameras to vision systems. In terms of layout, there are also two methods: traditional frame-mounted camera methods and industrial robot-mounted camera methods.

The improvement in system performance is mainly reflected in:The ability to overcome the problems of difficult extraction of laser stripes and unclear images caused by high-temperature radiation.The ability to improve the accuracy of extracting feature points of metal components and effectively remove the impact of the factory environment.The capability of adopting different algorithms to improve the operating speed of the system.The capacity to adopt feasible and effective protection equipment to reduce the influence of high temperatures, dust and other factors on equipment life.The ability to develop a reasonable neural network to detect the surface defects of the measurement target.

Table 8 compares the research results of multi-view stereo vision technology in terms of the measurement target, measurement equipment and key technologies.

## 4. Discussion and Comparative Analysis

This section compares the measurement values of high-temperature metal components from the two system measurement principles and detection target to provide a reference for relevant researchers.

1.Analysis of the differences in measurement principles:

The basic measuring principle of the time-of-flight ranging method is relatively simple. Data information can be obtained on the basis of the correlation between the laser transmitting module and the receiving module, and most of the data are suitable for remote measurement. The interference of ambient light can be reduced in the measurement of high-temperature objects. However, due to the fast propagation speed of light, the system requires a very precise time measurement device, so the cost is higher than that of laser triangulation. Commonly used equipment for time-of-flight measurement includes laser rangefinders, motion control platforms, and scanning devices.

The characteristics of laser triangulation are obvious. Its measurement accuracy is affected by the measurement distance; the greater the distance is, the greater the error, so it is often used for close-range high-precision measurements. The system is affected by the thermal radiation of high-temperature objects and the measurement of ambient light, but laser triangulation does not require sophisticated measurement equipment, so the cost is low. Commonly used equipment includes industrial cameras, laser transmitters and filters.

Multi-view stereo vision technology is an important direction of machine vision. Arranging multiple cameras has the advantages of increasing the measurement angle of high-temperature objects, solving the problem of environmental occlusion, and accurately obtaining 3D depth information of an object. However, the difficulty lies in the system calibration, feature extraction and stereo matching. It is necessary to accurately obtain the internal and external parameters of a camera, which requires a high matching accuracy of each camera image. The main equipment for measuring high-temperature metal components with multi-eye stereo vision includes CCD cameras (line array/area array), DLP projectors, filters and a data processing computer.

Table 9 compares and analyzes the advantages and disadvantages of different technologies from the perspective of measurement methods.

2.Analysis of different measurement objects and tasks:

To detect surface defects on high-temperature metal components, the current methods include laser triangulation and multi-eye stereo vision. It is necessary to overcome the influence of heat radiation from high-temperature metal components before successfully realizing the inspection task. The common method is to add filters. In defect detection, the main task is to use industrial cameras and other devices to obtain the 3D point cloud data or image data of a target, and the recognition task is realized through subsequent algorithms. Recognition algorithms often use deep learning for better performance, but deep learning requires a large amount of data for training to improve the success rate of defect detection. At present, defect detection is more concentrated on surface crack defect detection, but defects such as inclusions in the slab are detected by removing some surface materials. This reduces the detection efficiency, and the detection accuracy is not high. Therefore, achieving a more effective detection method for internal defects and developing a more comprehensive defect recognition algorithm remain the focuses of further research.

Cylindrical shell parts are the most common metal components in industrial production, and the methods used to measure them involve laser scanning measurements and multi-eye stereo vision measurements. The measuring instruments that use the TOF principle obtain the coordinates of metal feature points according to the measured distance information, which is used to reconstruct the high-temperature metal components and measure the diameter and length of the metal components. A laser rangefinder is installed on a special scanning device to obtain complete data through continuous movement. The structure of the scanning platform is complex; if the scanning device is not used, the angle of view is blocked, and the data cannot be collected effectively. In addition, precise control of the movement posture of the scanning platform can expand the application range of the measuring device. When a scanning device is not used, it is necessary to determine the installation position of the laser rangefinder according to different measurement targets. The measuring distance of the TOF principle can reach 15 m, which can effectively reduce the influence of high temperature on the measurement of metal components. When laser triangulation is used, a motor drives a laser transmitter to scan metal components, and a CCD camera completes the image collection task. In addition to the influence of thermal radiation on image quality, the error source of a system lies in the influence of the lens distortion during the calibration process of a CCD camera. The current research includes camera calibration, and the PSO algorithm is used to optimize the calibration results to improve system accuracy. In the measurement of cylindrical shell parts by multi-vision stereo systems, not only the common challenges faced by the measurement of high-temperature metal components but also the parameter calculation of the CCD camera should be considered. With a large number of cameras in the multi-vision system, it is more difficult to solve the internal, external and distortion parameters. Therefore, researchers have not ended their research on the calibration of multi-vision cameras. Stereo vision technology takes feature points as the basic unit of detection, and the measurement of a large cylindrical shell requires considerable data for processing. In recent years, some researchers have improved their methods by replacing characteristic points with characteristic lines for detection, which greatly improves the speed of measurement. However, the measurement accuracy of this method needs to be tested on different metal components.

Regardless of which principle is used to measure high-temperature forgings of other shapes, the algorithm is required to separate the forgings from the measuring environment. The methods currently used include angle constraints, distance constraints, hierarchical clustering analysis, and image segmentation algorithms. Only by removing irrelevant point cloud data can the subsequent measurement tasks be further completed. When measuring metal components with a temperature of 1000 °C based on the principle of laser triangulation, it is difficult to extract the laser profile because the metal components emit strong bright yellow light. Stereoscopic vision researchers have studied how to improve the quality of images at high temperatures. Since the deflection of visible light causes measurement errors, the use of an infrared thermal imaging camera to build a binocular vision system reduces the relative errors and overcomes the effect of high-temperature radiation to a certain extent. The integration time of the camera affects the image quality, and an adaptive exposure time control model is established to obtain an ideal grayscale image. The proposed combined filtering technology can also directly improve the image quality of high-temperature metal components. In addition, high-temperature radiation has a greater impact on the measurement of metal components in phase measurement profilometry (PMP), and it is impossible to obtain clear and deformed graphic images. This problem can be effectively solved by separating the R, G, and B channels of an image and then performing a phase operation under the gray image of the B channel. With the continuous advancement of robotics technology, stereo vision technology has gradually changed from equipment that is separately installed in factory environments for measurement to robots installed in factory environments for measurement. Robots can move more freely, which solves the problem of obstructed viewing angles. By designing the motion trajectory of a robot, the exposure time of the equipment in a high-temperature environment can be reduced, and the service life of the equipment can be prolonged. Vibrations of the factory production line affect the measurement accuracy of robot vision systems, and measurements need to be taken to compensate for the vibrations. In terms of improving measurement efficiency, CPUs and GPUs can be used to process data in parallel. Time-division multiplexing triggers can be used to make full use of the bandwidth of Gigabit Ethernet and can increase the measurement speed. For example, high-temperature metal components with complex surfaces, such as turbine blades and automobile front bridges, can be measured with the principle of laser triangulation and the principle of multi-vision stereo vision. To obtain comprehensive surface information from metal components, a large number of laser transmitters and industrial cameras are required when laser triangulation is used (the number used in the current study is eight). The movement of a mobile platform can obtain information on the third coordinate axis. During the measurement of complex metal components, to reduce the equipment damage caused by high temperatures and dusty environments, the system installs cooling devices, dust removal devices, protective plates and other protective devices.

Table 10 compares different measurement methods from the perspective of measurement objects.

## 5. Future Development Trends

The fourth revolution of science and technology provides a new driving force for the development of high-temperature metal component 3D size and morphology measurements. The accuracy and speed of measurement systems will continue to be improved, and miniaturized measurement systems, such as the low-depth camera systems of mobile phones [105,106,107,108,109,110], will continue to be produced. The following is an analysis of and outlook on the future development trends of the measurement of high-temperature metal components with stereo vision:

### 5.1. Fusion of Multi-Modal Image Data

With the continuing development of deep learning technology and the wide application of infrared sensors, a multi-modal measurement system that combines deep learning, camera image data and infrared image data will be one of the development directions of high-temperature metal component measurements. The powerful function of deep learning in image feature recognition can reduce the occurrence of false and missed defects. Existing equipment still cannot achieve the direct detection of internal defects in metal components and can only perform detection after removing the metal surface. Infrared technology can provide a solution to this problem. In addition to the 3D reconstruction of metal components, 2D image data often contain more original information. Whether image data can be used more effectively in the measurement of high-temperature metal components is a breakthrough direction for researchers. Multi-modal data fusion can be used to obtain more comprehensive information from high-temperature metal components and improve the measurement accuracy of the size and surface morphology.

### 5.2. Improvements in Various Hardware Equipment Adopted by a System

In the process of measuring high-temperature metal components, the performance of the equipment determines the accuracy of the measurement results. If a laser transmitter can emit finer laser stripes, the robustness of the system can be greatly improved, more comprehensive point cloud data can be obtained, the measurement of complex curved surfaces can be achieved, and even smaller defects can be detected. By improving the sensitivity and exposure of industrial cameras, higher-quality image data can be obtained in high-temperature environments, and the measurement accuracy can be improved directly. In the development of data processing hardware, the use of an RTX3090 graphics card will shorten the measurement time. The progress of MEMS makes it possible to integrate the devices of a measurement system. More miniaturized and mobile hardware devices will become more popular with users and will occupy a larger market.

### 5.3. Automatic Control Technology

Looking back at the development of measurement systems for high-temperature metal components, system calibration is a bottleneck that plagues equipment performance. Directly arranging stereo vision devices at a test site will not only cause viewing angle occlusion problems but also affect the acquisition of point cloud data. Automatic control technology can accurately control the camera position and pose, reduce calibration errors and reduce the workload of manual operation in the early stage. However, there is still a lack of breakthroughs in research on automation equipment.

### 5.4. Intelligent Development of Algorithms

Researchers have applied different algorithms to the 3D measurement of high-temperature metal components and have also achieved impressive experimental results. Deep learning algorithms have been applied to defect detection, and achieved good detection results [111,112]. As an important development direction of computer science, artificial intelligence (AI) has also experienced a rapid increase in development internationally in recent years. Researchers have already applied deep learning algorithms to surface defect detection on high-temperature metal components. The development of more intelligent algorithms is a future research trend.

## 6. Conclusions

In this paper, the measurement technology used in determining the 3D size and morphology of high-temperature metal components over the past decades was summarized. The advantages of laser scanning technology and the multi-view stereo vision measurement method in the measurement of high-temperature metal components were analyzed. According to the different measurement objects, a comparative analysis of various technologies can provide some reference for researchers engaged in the 3D measurement of high-temperature metal components.

Due to the influence of high-temperature environments, other traditional detection methods, such as contact measurement, often cannot meet the production needs of enterprises and cannot detect quality problems in production in time, thus causing the loss of economic benefits. In response to this practical problem, researchers from various countries have developed measurement systems to determine the size and morphology of high-temperature metal components and adopted a series of measures to reduce the influence of factors, such as high-temperature radiation, vibration, and dust erosion, which interfere with detection. In addition, continuous breakthroughs have been achieved in system measurement accuracy and speed, which have greatly promoted the improvement in high-temperature metal component detection technology and contributed to the improvement in the processing quality of metal components.

## Figures and Tables

**Figure 1 sensors-21-04680-f001:**
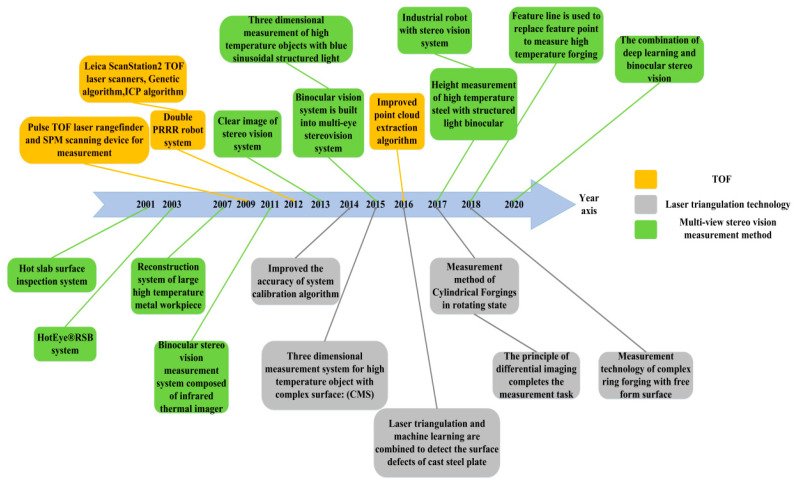
Timeline of technological development.

**Figure 2 sensors-21-04680-f002:**
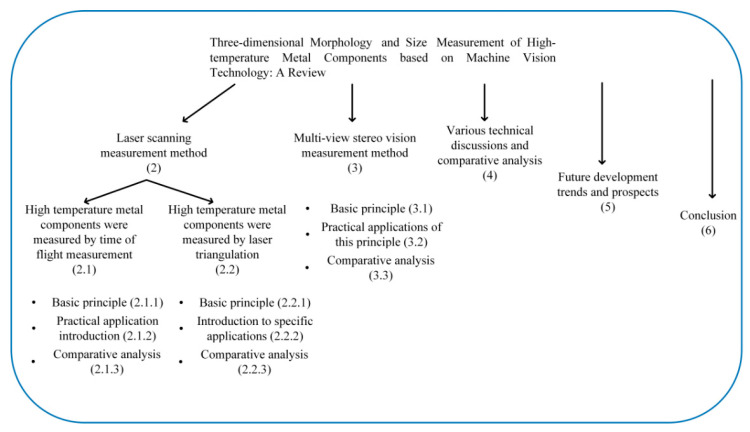
The frame structure of the article.

**Figure 3 sensors-21-04680-f003:**
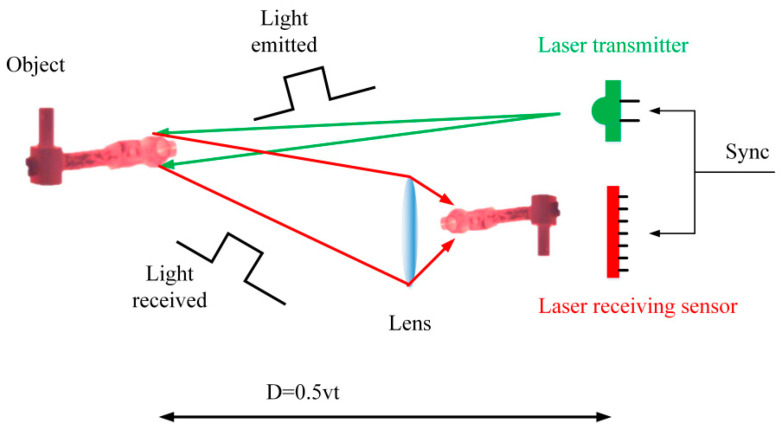
The basic principle of TOF.

**Figure 4 sensors-21-04680-f004:**
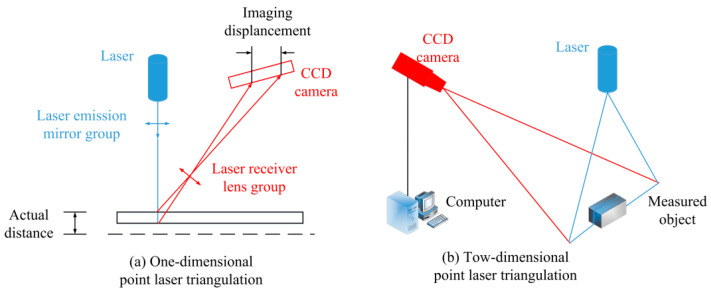
Basic principle of laser triangulation.

**Figure 5 sensors-21-04680-f005:**
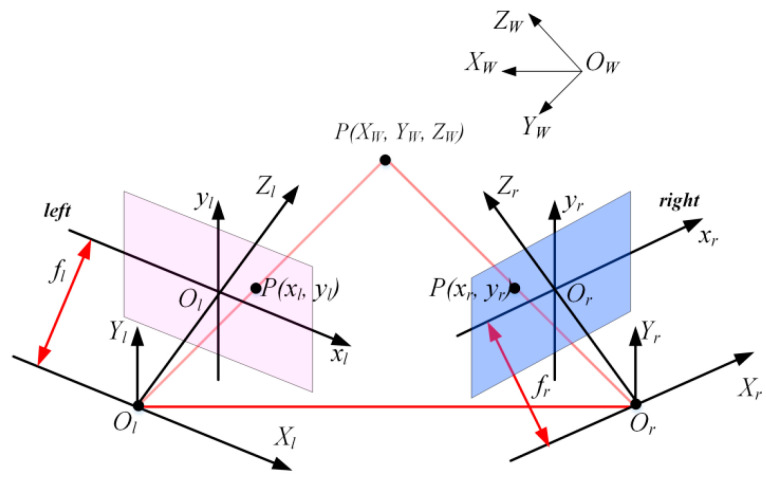
Principle of binocular stereo vision.

**Figure 6 sensors-21-04680-f006:**
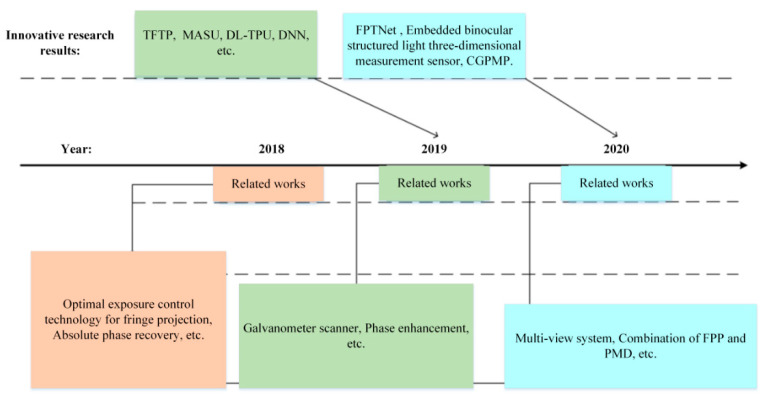
Raster projection technology development.

**Figure 7 sensors-21-04680-f007:**
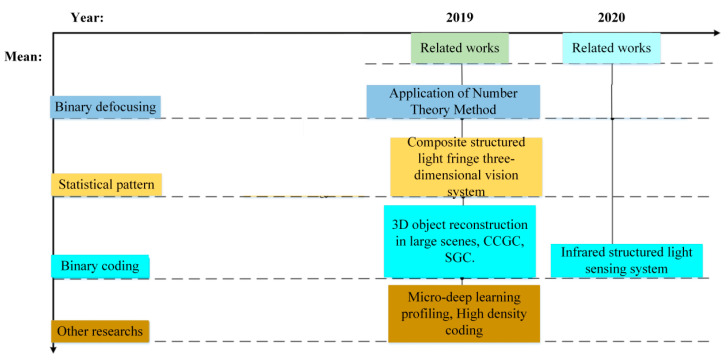
Technology timeline.

**Table 1 sensors-21-04680-t001:** Comparison of the actual application of TOF.

Ref.	Detection Target	Hardware Equipment	Key Technologies and Algorithms	Measurement Accuracy	Efficiency and Robustness
[14]	Shaft parts	3D scanner, Laser rangefinder	Cooling protective shell, Wide measurement perspective	All not given	Medium
[15]	Crankshaft	TOF laser rangefinder, SPM scanning device	Stable scanning system, Measurement self-adaptability	4.5 mm	High
[16]	Cylindrical metal housing	PRRR robot, Laser sensor	High precision of scanning system, Optimization of motion parameters	Measurement error is less than 0.235% and 0.205%	Medium
[17]	Cylindrical metal housing	Laser scanner	3D segmentation, RANSAC algorithm, ICP algorithm	≤8 mm.	Low
[18]	Large forgings	LMS100 radar, GT-400-SG motion control card	Three-step extraction algorithm, 3D Delaunay triangulation algorithm	2%.	Low

**Table 2 sensors-21-04680-t002:** Comparison of measurement quality of laser triangulation.

Ref.	Detection Target	Hardware Equipment	Key Technologies and Algorithms	Measurement Accuracy	Efficiency and Robustness
[25]	Cylindrical forgings	MV-VE078SM/SC camera, MGL-III laser transmitter	PSO algorithm	Measurement error is less than 1 mm.	High
[26]	Cylindrical forgings under rotation	VZ-1000 3D laser scanner	Least square method, Coordinate rotation processing	Measurement error is less than 6 mm.	Medium
[27]	Complex ring forgings	VQ-180 2D laser scanner	Topological embedding mapping, Topological differential theory	1.74 mm.	High
[28]	Complicated surface parts such as turbine blades	Mobile platform, Point laser transmitter	Combine eight sensors, Custom meshing algorithm	Length measurement error range 0.1 mm.	High
[29]	Surface defects of cast steel plate	SL-405-35-S-C-15.0 laser transmitter, UI-3370CPs IDS camera	Center of mass peak detection algorithm, Deep learning algorithm.	The system performs defect detection in the steel mill.	Medium
[30]	Large hot forging	camera.	Principle of differential imaging	1 mm.	High
[31]	Hot rolled metal component	CALIPRI RCX system	Rotating scanning technology	All not given	Medium

**Table 3 sensors-21-04680-t003:** Sinusoidal phase encoding.

Method	Ref.	Hardware Equipment	Number of Cameras	What Problem to Solve
Sinusoidal phase encoding	[49]	Camera (GS3-U3-23S6M), Projector (4500)	1	Automatically control exposure
	[50]	DMD projector	1	ERDF
	[51]	Camera (UI-3250CP-M-GL), Projector (PDC03)	1	Structured light projection system using MEMS.
	[52]	Camera (Point Grey Chameleon3), Projector (4500)	1	A phase-enhanced encoding method
	[53]	Cameras (acA1300- 200), Projector (PRO4500)	2	GPU parallel computing saves time.
	[54]	Camera (MER-131-210U3M-L), Projector (6500).	1	Improve the accuracy of absolute phase extraction.
	[55]	Camera (ECO445CVGE), Projector (4500)	1	3D measurement of complex reflectivity
	[56]	Camera (GC640C), Projector (XJ-M140)	1	Real-time calculation of phase.
	[57]	Camera (504C), Projector (4500)	1	Multi-target recognition and reconstruction
	[58]	SWIR camera	1	Low light measurement
	[56]	All not given	All not given	(GL) coding techniques, FPP and PMD techniques
	[60]	Mechanical projector	1	TFTP
	[61]	Computer (I7-8700)	1	MASU
	[62,63]	Camera (acA640-750), Projector (4500Pro)	1	DL-TPU
	[64]	i3-8100 processor	All not given	Convolutional neural network
	[65]	PC (i5-7500H)	1	Neural network
	[66]	Computer (1226 v3 CPU/K2200GPU)	4	Achieve 360° measurement
	[67]	Projector (DLP6500), Camera (acA800-510)	1	FPTNet
	[68]	Projector (4500), Camera (model: MER-301-125U3M)	2	embedded binocular structured light 3D measurement sensor
	[69]	Camera (GEV-B1610M- SC000), DLP (PLED-W200).	1	The effect of noise on accuracy
	[70]	Camera (model:22.6.3.2)	1	Reduce distortion error, system error
	[71]	LED projector	1	Improved system calibration accuracy
	[72]	Projector (4500), Camera (131-210U3M),	2	A new 3D matching frame
	[73]	Camera (Ace-1600gm), Projector (DELL M115HD)	1	Improved hybrid calibration method of the system

**Table 4 sensors-21-04680-t004:** Binary defocusing.

Method	Ref.	Hardware Equipment	Number of Cameras	What Problem to Solve
Binary defocusing	[74]	Camera (V611), XGA resolution (1024 × 768) DMD	1	Solve the problem of high-frequency phase unwrapping

**Table 5 sensors-21-04680-t005:** Statistical pattern.

Method	Ref.	Hardware Equipment	Number of Cameras	What Problem to Solve
Statistical pattern	[75]	Cameras (V611), Digital micro-mirror device (DMD)	2	Composite structured light stripes for 3D measurement

**Table 6 sensors-21-04680-t006:** Binary coding.

Method	Ref.	Hardware Equipment	Number of Cameras	What Problem to Solve
Binary coding	[76]	Camera (acA1920-155), Projector (MP-CL1A)	1	Large-scale scene 3D reconstruction technology
	[78]	Projector (4500), Camera (Mini UX50)	1	Cyclic complementary gray code
	[79]	Projector (4500), Camera (Mini UX100)	1	Reduced edge phase unwrapping error of codeword
	[80]	Camera (FLIR BFS-U3-04S2M), Projector (TI-DLP4500)	1	Dynamic infrared structured light sensing system

**Table 7 sensors-21-04680-t007:** Other works.

Method	Ref.	Hardware Equipment	Number of Cameras	What Problem to Solve
Other works	[81]	Projector (4100), Camera (V611)	1	Neural network for phase calculation
	[82]	All not given	All not given	Realize structured light high-density coding

**Table 8 sensors-21-04680-t008:** Application and comparison of multi-view stereo vision.

Ref.	Detection Target	Hardware Equipment	Key Technologies and Algorithms	Measurement Accuracy	Efficiency and Robustness
[83]	Metal slab	Scan camera	Cone holography	90%	Medium
[84]	Hot rolled metal component	Sensor, Computer module	On-line detection technology	±0.1 mm	Medium
[85]	Cylindrical metal component	Binocular vision system	Use infrared light for optical calibration	±5 mm	Medium
[86]	components in wind tunnel	Thermal Imager	BP artificial neural network	0.4 mm	High
[87]	Long shaft forgings	Camera (31BU03)	Section lines extract feature points, Euclidean distance algorithm	2.5–3.5 mm	Medium
[92]	Large forgings	Camera (ES4020)	Morphological operation, 3D matching algorithm	0.7%	Medium
[93]	Large forgings	Camera (ES4020)	ISNR compensation technology	0.32 mm,	High
[94]	Large forging	Camera (SVS11002)	Multi-eye vision system	0.10%	High
[95]	Forgings	Camera	Channel separation technique	Relative error is less than 1:1000	Medium
[96]	Steel	Camera (MER-030-120UC)	Combined filtering, CPU/GPU	2 mm	Medium
[97,98,99]	Forgings	IRB1600 industrial robot	Infrared cut filter, Path optimization algorithm	0.28mm	High
[100]	Automobile front axle section	Air cooling device, Camera.	Three-frequency four-step phase shifting method,	Online measurement	High
[101]			Vibration error compensation algorithm		High
[102]	Large forgings	Camera (MER-132-30GC)	3D reconstruction technology of feature line	0.79%	High
[103,104]	Surface defects	Camera (MER-500-14GC-P)	Binocular vision system, Neural network	Use ACC to test system performance	Medium

**Table 9 sensors-21-04680-t009:** Comparison of measurement methods.

Method	Advantage	Disadvantage	Precision	Applicable Scenario
TOF	Principle is simple and reduces the interference of ambient light	High cost, low precision	Low	Remote measurement
Laser triangulation technology	Low cost	The larger the measurement distance, the greater the error, sensitive to external interference	Medium	Close-range and high-precision measurement
Multi-view stereo vision measurement method	Measurement Angle of view is many, highest precision	Principle is complex	High	Measurements where depth information is required

**Table 10 sensors-21-04680-t010:** Comparison of measurement objects.

Object of Measurement	Method	Technical Difficulties
High-temperature metal components	Laser triangulation, Multi-eye stereo vision	High-temperature thermal radiation effect, Fine internal and surface defects are difficult to identify
Cylindrical shell parts	Laser scanning measurements, Multi-eye stereo vision measurements	Large parts are difficult to measure, Scanning device has low accuracy, Lens distortion error
High-temperature metal components with complex surfaces	Laser triangulation, Multi-vision stereo vision	Complete surfaces are difficult to obtain, Occlusion exists in the measurement perspective

## Data Availability

No new data were created or analyzed in this study. Data sharing is not applicable to this article.

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
