# Peer review of "Three-Dimensional Morphology and Size Measurement of High-Temperature Metal Components Based on Machine Vision Technology: A Review"

_sensors, 2021, doi:10.3390/s21144680_

Round 1

Reviewer 1 Report

This paper reviewed the utilization of 3D camera in Metal components. 
However, I do not think the paper is a good organized for this paper only listed a lot of papers. The readers cannot get the historic improvement of the technology of using 3D camera in the field of meta components.

I give the writers the following suggestions for revising the paper.

1. Please list out the technology improvement by ages in Figure 2, because a review paper always show the historic of the technology to readers.

2. Write the cited paper number at the last of title of the figures when the figures are cited from other papers.

3. In the paper, there are too many figures repeat the binocular stereo vision system. The writers do not need show out every picture of the original paper.

Reviewer 2 Report

This paper reviewed the past and current research fields of the metal component size measurement within the high temperature condition. This paper is well organized and the applied industrial case studies well illustrated in detail. However, there are some minor comments for the authors,

  1. Some figures are unclear e.g., Figures 2 and 3.
  2. The measuring principle shall be reduced as they were all well-known.
  3. The structure of this paper is listed with the individual tabulated measuring technology. From the view of applications, a tabulated comparison between different measuring approach is practical for the readers to get insights into the advantages, limitations and drawbacks of each approaches.

Reviewer 3 Report

Most, if not all, images and figures present in this paper are directly taken from the cited references. As this presents an considerable case of plagiarism, I have to reject the publication of this article.

Round 2

Reviewer 3 Report

The authors provide an extensive literature review on vision-based measurement technologies applied to high temperature metal components. 

Based on previous comments and revisions, this reviewer agrees on publication as is.

Author Response

Sincerely thank you for your recognition of our work.